# Gastric Cancer with Peritoneal Metastases: Current Status and Prospects for Treatment

**DOI:** 10.3390/cancers15061777

**Published:** 2023-03-15

**Authors:** Israel Manzanedo, Fernando Pereira, Estíbalitz Pérez-Viejo, Ángel Serrano

**Affiliations:** 1Department of General and Digestive Surgery, Peritoneal Carcinomatosis Unit, Hospital of Fuenlabrada, 28943 Madrid, Spain; 2Department of Surgery, Rey Juan Carlos University (URJC), 28933 Madrid, Spain

**Keywords:** gastric cancer, peritoneal metastases, HIPEC, cytoreductive surgery

## Abstract

**Simple Summary:**

Peritoneal metastases from gastric cancer have a poor prognosis. Their presence usually indicates the end-stage of the disease without curative treatment. However, the treatment of these patients with cytoreductive surgery and intraperitoneal chemotherapy has achieved promising results in different studies that have been published in recent years. However, this treatment approach is controversial. This review aims to update those studies to clarify whether surgery and intraperitoneal chemotherapy are beneficial for these patients.

**Abstract:**

Gastric cancer (GC) has a poor prognostic and only one in four patients will have survived by 5 years after diagnosis. These poor results are due to the fact that most patients are diagnosed in advanced stages; peritoneal metastases (PM) are especially frequent and are difficult to treat. Currently, PM are considered a terminal stage of GC with a poor survival rate and are treated with palliative systemic chemotherapy. Since the beginning of the century, the treatment of PM from different origins has evolved; cytoreductive surgery (CRS) and hyperthermic intraperitoneal chemotherapy (HIPEC) have become the treatment of choice for many malignant diseases that affect the peritoneum. CRS and HIPEC have also been used for patients with GC and PM, achieving survival results that have never been achieved when using systemic chemotherapy alone. The use of HIPEC can even prevent the development of peritoneal recurrences in patients with locally advanced GC as adjuvant therapy, can reduce the volume of peritoneal disease as neoadjuvant therapy, and can control symptoms in a palliative setting. The aim of this review is to collate the current scientific evidence regarding the treatment of PM of GC origin with surgery and intraperitoneal chemotherapy.

## 1. Introduction

Gastric cancer (GC) has a very poor prognosis and represents the third most common cause of cancer-related deaths in the world, despite being the fifth most commonly diagnosed type of cancer [1,2]. Only one in four patients will survive up to 5 years [3,4]. The majority of patients are diagnosed in advanced stages; peritoneal metastases (PM) are especially frequent and are present in 15–30% of patients at diagnosis [5,6,7]. Moreover, 40–60% of patients treated with a curative gastrectomy and systemic chemotherapy program will develop a peritoneal relapse as the only site of recurrence [6,7,8].

The treatment of GC depends on its stage, as determined by the American Joint Committee on Cancer (AJCC) tumor-node metastases (TNM) staging system, updated in 2018 [9]. Early and locally advanced GC can be treated with curative intent by surgery with or without systemic chemotherapy, depending on each case, and the 5-year survival rate for these cases can reach 55% [5,10].

However, the standard treatment for metastatic GC is still palliative systemic chemotherapy, with a poor prognosis and median overall survival (OS) of 6 months [3,11,12]. Advances in systemic treatments have improved the survival rates for patients with stage-4 disease, being able to achieve survival periods of up to 8–14 months in selected cases [11,13,14,15]; however, when PM are present, these results are worse, the patients survive less than 6 months, and there are no survivors after 5 years [3,11,16,17]. Even the patients with positive peritoneal cytology (microscopic PM) and without macroscopic nodes are classified at stage 4 and have a similar prognosis and the same treatment as the other metastatic GC patients [4].

Since the 1990s, the knowledge of PM development has made great progress; this has changed the classical belief that peritoneal implants constitute systemic disease, this now being considered a locally disseminated disease [18]. The treatment of PM by combining systemic chemotherapy, cytoreductive surgery (CRS), and hyperthermic intraperitoneal chemotherapy (HIPEC) has achieved encouraging results for different tumors that affect the peritoneum. CRS can remove the macroscopic disease (peritoneal nodules) through different peritonectomy procedures and visceral resections, while the microscopical disease is treated locally with intraperitoneal chemotherapy [19,20].

Currently, this multimodal treatment is the standard of care for pseudomyxoma peritonei and peritoneal mesothelioma [21,22]. In selected patients with PM due to colorectal cancer, CRS is an accepted treatment; the addition of HIPEC has been questioned after the negative results published from the PRODIGE-7 trial, but there are ongoing studies regarding its validation [23,24]. CRS, combined with systemic chemotherapy, is the standard treatment for advanced ovarian cancer, but evidence of the efficacy of HIPEC has increased in recent years regarding this disease [25]. This multimodal treatment has been also used to treat PM from GC, with encouraging results, according to different retrospective studies and several comparative studies [3,4,5,6,7,17,26,27,28,29].

We have made this review to provide an update on the use of CRS and HIPEC in patients with GC.

## 2. Prevention of Development of PM in Locally Advanced GC

The peritoneal cavity is the most common site of tumor recurrence after radical gastrectomy with D2 lymphadenectomy in cases of locally advanced GC. Currently, the standard perioperative complementary therapy is systemic chemotherapy. In spite of correct treatment, peritoneal relapses will still occur in about 40% of patients [6,8,11]. Systemic chemotherapy has serious difficulties in penetrating the blood–peritoneal barrier and accessing the peritoneum; for this reason, other types of treatments have been sought that could act more effectively on the peritoneum [5]. If the drug is administered directly into the peritoneal cavity, the blood–peritoneal barrier will not allow it to penetrate into the bloodstream, at least partially; therefore, we can achieve high local concentrations of the drug without the side effects that are derived from its systemic administration.

The use of adjuvant HIPEC after a curative gastrectomy in patients with locally advanced GC, without PM, can prevent a peritoneal relapse. There are numerous studies showing the effectiveness of HIPEC as a prophylactic treatment (Table 1).

In 2001, Yonemura et al. randomized 139 patients into three treatment groups (surgery alone, surgery plus normothermic intraperitoneal chemotherapy (NIPEC), and surgery with HIPEC). The HIPEC group showed the best results, with a 5-year OS of 61%, vs. 44% and 42% in the other two groups (*p* = 0.021) [30].

In 2020, Xie et al. published a comparative study of 113 patients with locally advanced GC. In total, 51 patients were treated with HIPEC and systemic chemotherapy, while 62 patients were treated with systemic chemotherapy alone. The OS rate was significantly higher in the HIPEC group (*p* = 0.044), due to the decrease in peritoneal recurrence, and was significantly lower in the HIPEC group (3.9% vs. 17.7%) [31].

Moreover, there are several meta-analyses that confirm the survival benefit of adjuvant HIPEC [5,28,32]. In 2017, Desiderio et al. published a meta-analysis of 11 clinical trials and 21 observational studies, involving more than 2000 patients. The 5-year OS for patients treated with adjuvant HIPEC was better than that of the patients treated with surgery alone (RR = 0.82, *p* = 0.01) [28]. Recently, in 2022, Zhang et al. published a new meta-analysis; they analyzed 8 randomized trials and 5 non-randomized trials, enrolling 1201 patients with locally advanced GC without PM. The OS rate for those patients treated with surgery and prophylactic HIPEC was better than patients without HIPEC because the HIPEC treatment prevented peritoneal metastases (RR, 0.35; *p* < 0.0001), without increasing the morbidity (RR, 1.15; *p* = 1.51) [5].

**Table 1 cancers-15-01777-t001:** Prophylactic hyperthermic intraperitoneal chemotherapy (HIPEC) for locally advanced gastric cancer without peritoneal metastases.

Author (Year)	Patient Number	Cytostatic	Morbidity	Survival Results
Fujimoto (1999) [33]	71 (surgery + HIPEC) vs. 70 (surgery alone)	MMC	2/71 vs. 2/70	4-year OS: 76% vs. 58%
Yonemura (2001) [30]	48 (surgery + HIPEC) vs. 44 (surgery + NIPEC) vs. 47 (surgery alone)	MMC + Cisplatin	19% vs. 14% vs. 19% (mortality: 4% vs. 0% vs. 4%)	5-year OS: 61% vs. 44% vs. 42%
Zhu (2006) [34]	41 (surgery + HIPEC) vs. 53 (surgery alone)	MMC+ Cisplatin	23% vs. 12%	4-year OS: 70% vs. 52%
Cui (2014) [35]	48 (surgery) vs. 48 (NAC + surgery) vs. 48 (surgery + HIPEC) vs. 48 (NAC + surgery + HIPEC)	Cisplatin	No differences	3-year OS: 35% vs. 62% vs. 58% vs. 75%
Desiderio (2017) [28]	731 (surgery + HIPEC) vs. 1079 (surgery alone)	MMC, cisplatin, etoposide	Higher risk in HIPEC group (RR = 2.17)	5-year OS better in HIPEC (RR = 0.82)
Xie (2020) [31]	51 (surgery + HIPEC) vs. 62 (surgery alone)	Cisplatin	No differences in severe morbidity (6% vs. 6%)	3-year OS 69% vs. 66% (*p* = 0.04)
Zhang (2022) [5]	561 (surgery + HIPEC) vs. 640 (surgery alone)	Cisplatin, MMC, etoposide	No differences (RR = 1.15)	3-year OS better in HIPEC (RR = 0.63)

HIPEC: Hyperthermic intraperitoneal chemotherapy; MMC: mitomycin C; OS: overall survival; NIPEC: normothermic intraperitoneal chemotherapy; NAC: neoadjuvant chemotherapy.

Despite these results, HIPEC has not been established as an adjuvant therapy to prevent peritoneal recurrence in the Western world. Most of the studies in this context have been published in Asia; this fact generates the eternal scientific dilemma of whether the results obtained in Asian patients can be extrapolated to Western patients. The GASTRICHIP trial is a European study (French, with the collaboration of a few Spanish centers) that evaluates the effect of HIPEC as adjuvant therapy in locally advanced GC [36]. The recruitment has concluded, and the results will be published soon. Indeed, the safety results of the first 200 patients enrolled in the trial were presented at the second Congress of the International Society for the Study of Pleura and Peritoneum, 2021, and there were no differences in morbidity seen between the groups [37].

## 3. Treatment of PM of GC Origin

The treatment of PM of GC origin is complex and aggressive, with limited results; therefore, the correct identification of patients is essential to choose the appropriate treatment for each one. When a patient is diagnosed with GC with PM, it is essential to assess the extent of the disease. The volume of peritoneal disease must be carefully evaluated to identify those patients that are potentially curable; diagnostic laparoscopy is required to establish this peritoneal disease extension using the peritoneal cancer index (PCI) [38]. The abdominal cavity is divided into 13 anatomical regions, and each area is given a value from 0 to 3, based on the volume of peritoneal disease in that region; therefore, a numerical value is set, ranging from 0 to 39.

The value of PCI is essential when selecting patients for CRS treatment with HIPEC because the volume of peritoneal disease is an independent prognostic factor. In addition, complete cytoreduction is necessary to obtain good survival results. PCI and the grade of completeness of cytoreduction are related factors, assessed via the completeness cytoreduction score (CCS) according to the residual tumor, this being CCS-0 for no macroscopic residue, CCS-1 for the macroscopic residue of < 2.5 mm, CCS-2 for 2.5 mm to 2.5 cm tumoral residue and CCS-3 with a residual tumor of > 2.5 cm [39]); the possibility of achieving cytoreduction at CCS-0 is inversely proportional to PCI. Several studies in the last few years have recommended a PCI limit to propose curative treatment [7,17,40].

In 2010, Glehen et al. recommended a PCI limit of 12 because no patient with PCI > 12 survived in their analysis of 159 patients with GC and PM. They also observed an improvement in survival if the CRS value was CCS-0 [17]. Since this study was published, patients with a PCI higher than 12 are usually rejected for CRS and HIPEC in expert centers. The current trend is to be even more restrictive in terms of the PCI limit; with a PCI lower than 7 and complete cytoreduction (CCS-0), the possibility of a cure is real. In 2016, Chia et al. published a median OS of 26.4 months for patients with a PCI of < 7, vs. 10.9 months in patients with a PCI of ≥ 7 [40]. In the Spanish Registry, published in 2019, in an analysis of 88 patients, patients with a PCI of < 7 had a median OS of 26.1 months (5-year OS of 46.8%) while those patients with a PCI of ≥ 7 had a median OS of 18.9 months (5-year OS of 0.0%) [7]. The German Registry in 2020, with 235 patients, also showed better OS with a PCI of < 7 (a median OS of 18 months for patients with a PCI of 0–6, 12 months for patients with a PCI of 7–15, and 5 months for patients with a PCI of 16–39) [29].

Microscopic peritoneal disease (positive cytology) without macroscopic PM is considered a stage-4 disease, with a similar prognosis to those with visible PM. Different studies have demonstrated that treatment with HIPEC in these cases obtains encouraging survival results. In 2018, Rihuete et al. showed a 5-year OS of higher than 60% in patients with positive cytology [41]. In our Spanish Registry, those patients with isolated positive peritoneal cytology (without macroscopic disease) did not reach their median OS, with a long follow-up [7].

Patients with a potentially resectable PM should receive a multimodal treatment. Perioperative systemic chemotherapy is the standard treatment in Western countries [42,43]. Since 2019, the FLOT scheme (fluorouracil, leucovorin, oxaliplatin, and docetaxel) is the standard of care for locally advanced GC and for oligometastatic patients; it consists of 8 perioperative cycles (4 preoperative and 4 postoperative) [43].

Different ways have been employed to administer intraperitoneal chemotherapy as a neoadjuvant treatment to increase the number of complete cytoreductions because intraperitoneal chemotherapy can reduce the load of peritoneal disease.

Laparoscopic HIPEC has been evaluated as a neoadjuvant therapy. In 2017, Yonemura et al. showed a significant PCI decrease after the administration of 2 cycles of laparoscopic neoadjuvant HIPEC (docetaxel-cisplatin combination) in 53 patients, reducing the median PCI from 14.2 to 11.8 and turning positive cytology into negative cytology in 68% of patients [44]; also in 2017, Badgwell et al. reported the total disappearance of peritoneal disease in 7 of 19 patients treated with laparoscopic HIPEC (a mitomycin-C and cisplatin combination) [45].

Yonemura developed a neoadjuvant treatment with combined intraperitoneal and systemic chemotherapy (NIPS) in 2006 [46]. This treatment combines the intraperitoneal administration of chemotherapy, via a catheter introduced into the peritoneal cavity, with intravenous chemotherapy. In 2012, Yonemura et al. published the results of 96 patients treated by NIPS; 70% of these patients achieved a CCS-0 cytoreduction and 36.8% had a complete response in terms of pathologic analysis [47]. Different studies have reported similar results with different NIPS regimens [48,49,50]. In 2019, Hao et al. published the results of 69 patients treated with NIPS with docetaxel and cisplatin; they observed a downstaging of lymph node metastases after NIPS [49]. Recently, in 2021, Zhang et al. reported the results of a propensity score-matched analysis comparing 71 patients treated with NIPS vs. 71 treated with neoadjuvant systemic chemotherapy; NIPS yielded a better ascites and cytology response, a higher conversion resection rate and R0 resection rate, and better OS than systemic chemotherapy alone [50].

Nowadays, CRS and HIPEC represent a treatment for selected patients with GC and PM that has had the best survival results published (Table 2).

A complete CRS is essential to achieve good outcomes and must be carried out in experienced centers [7]. CRS must remove all macroscopic disease via gastrectomy, D2 lymphadenectomy, and peritonectomy, according to the established Sugarbaker techniques [19]. Despite removing all visible disease with surgery, microscopic disease must ideally be treated to prevent peritoneal relapse. HIPEC is used to treat this invisible disease, delivering a high local dose of chemotherapy into the peritoneum, with the synergism of the heat increasing the power of the drug against the tumoral cells [51].

Numerous retrospective and case-control studies have been published that obtained encouraging results for CRS with HIPEC [7,17,29,41,52]. In 2011, one randomized clinical trial was published wherein 68 patients were randomized to CRS with or without HIPEC; the patients treated with HIPEC (mitomycin C with cisplatin) had a significantly better survival rate (median OS of 11 months vs. 6.5 months, and 3-year survival rates of 5.9% vs. 0%) [26].

Most recently, in 2019, Bonnot et al. published the CYTO-CHIP study. This is a propensity score study that compared the outcomes of 180 patients treated with CRS and HIPEC with those of 97 patients treated with surgery alone. The best survival results were observed in the HIPEC group, with a median OS of 18.8 months vs. 12.1 months, and without differences in terms of morbidity or mortality [27].

The evidence has grown in recent years with the publication of several meta-analyses. In 2017, Desiderio et al. published a systematic review including 620 patients with GC and PM, of whom 289 were treated with CRS and HIPEC and 331 were treated without HIPEC. The median OS was increased in 4 months for the HIPEC group (11.1 months vs. 7.06, *p* < 0.001) [28].

Two meta-analyses have been published in 2022. Zhang et al. analyzed 691 patients diagnosed with PM of GC origin, of whom 384 were treated with CRS and HIPEC and 307 with CRS alone; their findings established a beneficial function for HIPEC, increasing the mean OS in 4.67 months without differences in morbidity [5]. Martins et al. also published a very interesting meta-analysis; they selected four non-randomized clinical trials and one clinical trial, analyzing a total of 475 patients with GC and PM, with 299 undergoing CRS and HIPEC and 176 undergoing CRS alone. They observed a significantly lower recurrence risk for CRS and HIPEC compared to CRS alone (RR = 0.23, *p* < 0.001); 1-, 3-, and 5-year OS rates were better in the HIPEC group, and no differences were found in the complication rates between the groups [6].

There is much controversy about the treatment of patients with more aggressive histological subtypes of GC, such as signet-ring cell tumors. This histological subtype is present in 30–40% of the GC cases diagnosed, and the patients frequently develop peritoneal disease. In the only randomized trial, published in 2011, 12% of patients in the HIPEC group had signet-ring cell tumors, vs. 0% in the non-HIPEC group, and the survival results were better in the HIPEC group [26]. In addition, the study published by Rihuete et al. showed that the presence of signet-ring cells was not a poor prognostic factor [41]. For these reasons, signet-ring cell tumors with PM do not have worse prognoses and therefore should not be ruled out for treatment with CRS and HIPEC.

Despite these results from the different studies, CRS and HIPEC are not considered the standard treatment regimen for patients with GC and PM. The absence of randomized clinical trials in the Western world is the cause of this fact. Currently, there are several Asian trials and two European phase-III studies that are ongoing (GASTRIPEC, NCT02158988 and PERISCOPE II, NCT03348150), which will evaluate the role of HIPEC in PM of GC origin [53]. The GASTRIPEC results were presented at the ESMO (European Society of Medical Oncology) and ESSO (European Society of Surgical Oncology) congresses in 2021; there was no difference in median OS with or without HIPEC, but when CRS was complete, a significant increase in median OS was observed. The 5-year OS was significantly higher in the HIPEC group (10% vs. 0%) and the authors did not observe any differences in morbidity or mortality; these results are pending publication [37,53]. The Dutch PERISCOPE II study has enrolled 106 patients and the estimated study completion date is October 2022 [52]. We hope that the future publication of these studies will provide us with more insights into this promising treatment.

All the results published to date show an improvement in median OS (except for the preliminary results of GASTRIPEC). This improvement may be considered modest since we are talking about only a few months (6 months in the Bonnot study and 5 months in the Yang clinical trial) [26,27]. However, what is really outstanding is their evidence of long-term survival with HIPEC, as highlighted by Badgwell in a recently published editorial, with 3- and 5-year survival rates that are three or four times higher than has been seen for treatment with CRS alone [37]. For example, in the GASTRIPEC trial, 5-year survival was at 10% in the HIPEC group and 0% in the non-HIPEC group [54], whereas in the CYTO-CHIP study by Bonnot, there was a 5-year OS of 20% in patients treated with HIPEC, compared to 6% in patients treated with CRS alone [27]. Therefore, long-term survival seems to be a variable that is more accurate than the median OS for assessing the improvement of survival rates in these patients.

Finally, there are still many unanswered questions about HIPEC as a delivery technique, the type of cytostatic to be used, the type of carrier solution, or the time of perfusion. Probably, a more international consensus should be promoted to try to standardize a technique that today is very heterogeneous; in fact, the international Delphi consensus, which was recently published [55], will contribute to standardizing these technical aspects.

After the above review and based on our own experience, we have developed an algorithm for the treatment of PM from GC (Figure 1).

## 4. Palliative Treatment

A high percentage of patients with GC will have unresectable disease and their treatment should be aimed at symptom control. Most symptoms are secondary to the development of ascites, which can cause pain, dyspnea, early satiety, or fatigue [11].

Laparoscopic HIPEC has been used to reduce ascites and improve the symptoms. Different studies have shown an improvement in the symptoms of 95% of patients. Facchiano et al. published a systematic review of 76 patients treated with palliative laparoscopic HIPEC; 95% of these patients showed ascites control without severe morbidity [56].

Pressurized intraperitoneal aerosol chemotherapy (PIPAC) is a new administration technique involving chemotherapy via laparoscopy. This therapy has been described for palliative patients with unresectable PM [57]. In 2019, Alyami et al. published a systematic review that included 838 patients with PM (185 from GC); the clinical response rate was 50% to 91% [58]. The same authors published a retrospective analysis in 2021 that included only patients with unresectable PM of GC origin. They reported 163 PIPACs in 42 patients; major complications were observed in 3% of patients, the OS was 19 months, and 6 patients (14%) were converted to CRS and HIPEC [59]. There are many ongoing studies on PIPAC when used as palliative treatment as well as neoadjuvant therapy to reduce the tumor burden and enable rescue CRS. PIPAC is approved in Europe but is not yet approved for use in the United States. The results of PIPAC are promising, but it is an expensive treatment that requires an invasive and repeated procedure [60]. Until now, PIPAC has not been compared with any other treatments (laparoscopic HIPEC, systemic chemotherapy plus biologic therapy, normothermic intraperitoneal chemotherapy in cycles), so it is still premature to draw too optimistic a conclusion about its results.

In summary, laparoscopic HIPEC and PIPAC are two good alternatives as palliative treatments in patients with unresectable PM. Both treatments are even being tested as conversion therapy to rescue patients in order to achieve complete cytoreduction. In the future, we will have more answers when the ongoing studies are completed (for example, the CHIMERA trial, NCT04597294 in the context of laparoscopic HIPEC, or trial PIPAC_VEROne, NCT05303714, for PIPAC).

New treatments based on molecular alterations, such as trastuzumab for HER2-positive patients or immunotherapy (nivolumab or pembrolizumab) in patients with microsatellite instability or PDL-1 overexpression, are being used in a palliative context with very good results. There are even studies underway using these drugs as neoadjuvant therapy, presenting surprising results. This immunotherapy may change the prognosis in highly specific cases and, in the future, we may need to rethink the treatment of patients in cases that today are considered palliative.

## 5. Conclusions

Patients diagnosed with GC have a high risk of developing PM, especially those patients with locally advanced GC. The use of adjuvant HIPEC may be beneficial for these patients and may prevent peritoneal relapse, improving survival rates. Those patients diagnosed with GC and PM should have their treatment managed in experienced centers. Patients with low-volume PM can be treated via neoadjuvant systemic chemotherapy, with CRS and HIPEC obtaining promising results, showing long-term survival rates never before seen in this cohort of patients and without more complications. Patients with high-volume PM can be treated via laparoscopic HIPEC or PIPAC to control their symptoms in a palliative setting. Despite the fact that the scientific evidence is growing, more studies in Western countries are necessary to corroborate the results of Asian studies. Immunotherapy and other new treatments may change the prognosis of selected patients; their combination with CRS and HIPEC in the future is, at present, a mystery. There is still a long way to go in terms of the treatment of GC with PM.

## Figures and Tables

**Figure 1 cancers-15-01777-f001:**
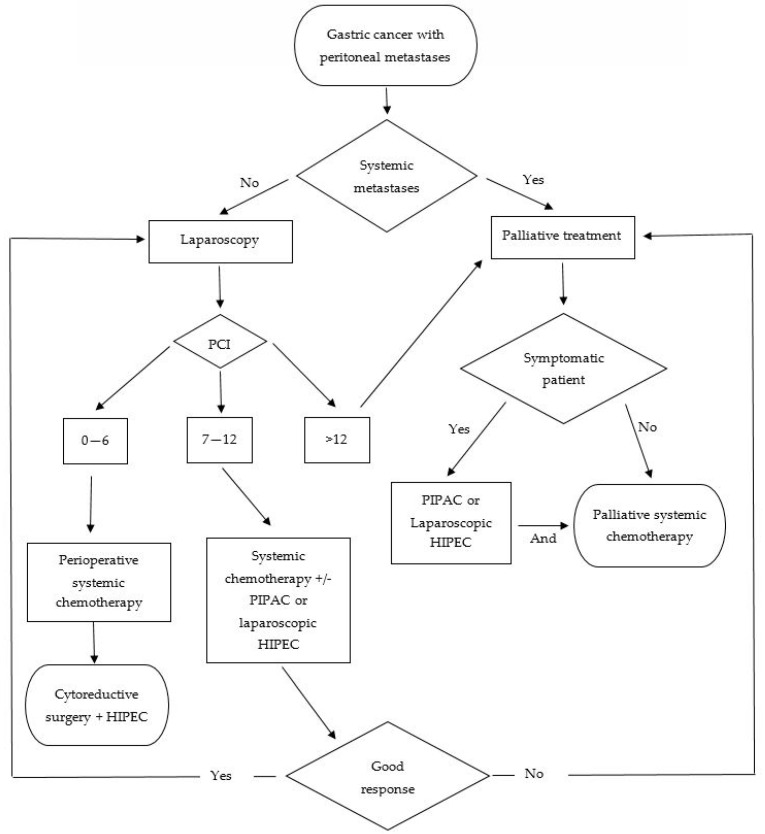
Therapeutic algorithm for gastric cancer with peritoneal metastases.

**Table 2 cancers-15-01777-t002:** Treatment with cytoreductive surgery (CRS) and the hyperthermic intraperitoneal chemotherapy (HIPEC) of peritoneal metastases of gastric cancer origin.

Author (Year)	Patients Number	Cytostatic	Morbidity	Survival Results
Glehen (2010) [17]	159 (CRS + HIPEC)	MMC, Cisplatin, Oxaliplatin	Grade 3–4: 27.8%. Mortality: 6.5%	Median OS: 9.2 months; 5-year OS: 13%
Yang (2011) [26]	34 (CRS + HIPEC) vs. 34 (CRS)	MMC + Cisplatin	14.7% vs. 11.7%. Mortality: 0%	Median OS: 11 vs. 6.5 months
Desiderio (2017) [28]	289 (CRS + HIPEC) vs. 331 (CRS)	MMC, Cisplatin, Etoposide	Higher risk in HIPEC group (RR = 2.15, *p* < 0.01)	Median OS: 11 vs. 7 months
Manzanedo (2019) [7]	88 (CRS + HIPEC)	Cisplatin, Doxorubicin, MMC, Oxaliplatin	Grade III-IV D-C: 31%. Mortality: 3.4%	Median OS: 21.2 months; 3-year OS: 30.9%
Bonnot (2019) [27]	180 (CRS + HIPEC) vs. 97 (CRS)	MMC, Cisplatin, Oxaliplatin	53.7% vs. 55.3%. Mortality: 7.4% vs. 10.1%	Median OS: 18.8 vs. 12.1 months; 5-year OS: 19.9% vs. 6.4%
Rau (2020) [29]	235 (CRS + HIPEC)	MMC, Cisplatin, Doxorubicin, Oxaliplatin	Grade III-IV D-C: 17%. Mortality: 5.1%	Median OS: 13 months; 5-year OS: 6%
Zhang (2022) [5]	384 (CRS + HIPEC) vs. 307 (CRS)	MMC, Cisplatin, Doxorubicin, Etoposide	No morbidity differences	Higher mean OS in HIPEC group (4.67 months)
Martins (2022) [6]	299 (CRS + HIPEC) vs. 176 (CRS)	MMC, Cisplatin, Etoposide	No morbidity differences	A 5-year OS, 3 times higher in HIPEC (RR = 3.25, *p* = 0.010)

CRS: Cytoreductive surgery; HIPEC: hyperthermic intraperitoneal chemotherapy; MMC: Mitomycin C; NA: not available; OS: overall survival; D-C: Dindo–Clavien classification.

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
