# Peer review of "Gastric Cancer with Peritoneal Metastases: Current Status and Prospects for Treatment"

_cancers, 2023, doi:10.3390/cancers15061777_

Round 1
Reviewer 1 Report (New Reviewer)
Gastric cancer with peritoneal metastases: Current status and prospects for treatment.
The subject is important even if no such recent progress had been observed.
The major limitation is that the paper really focus on cytoreductive surgery and HIPEC but is very limited for the rest of the treatment proposed including new drugs, new delivery solution, precision medicine identification (HER2, MSI, etc) , use of intra peritoneal drugs in ip. Procedure or PIPAC delivery solution.
For that reason the tittle of the paper had to be completely changed as :
The place of cytoreductive surgery and HIPEC in case of gastric peritoneal disease
By example for PIPAC procedure
Reference 56 had no interest for a review
But please include publication as:
In a recent paper published in 2021 Pressurized intraperitoneal aerosol chemotherapy (PIPAC) for unresectable peritoneal metastasis from gastric cancer.
Alyami M, Bonnot PE, Mercier F, Laplace N, Villeneuve L, Passot G, Bakrin N, Kepenekian V, Glehen O. Eur J Surg Oncol. 2021 Jan;47(1):123-127.
“Six (14.3%) patients became resectable during treatment and underwent curative intent CRS and HIPEC.”
This information can be find in different recent published papers – If authors read the literature they are 60 papers published on PIPAC and gastric cancer, a table had to be done with the potential use of PIPAC to change the resecability of gastric peritoneal metastasis.
Minor remarks:
Table 2, the paper by
Martins 2022 is not an original paper but a review of 5 priors published paper – if confirm please delate
Page 8
Line 262
“The results of PIPA are promising,”
Change for “The results of PIPAC are promising,”
Author Response
Please see the attachment

Reviewer 2 Report (New Reviewer)
Reading this paper, althought it is well structured, I think it is treated briefly.
I would suggest to integrate with data on the different histological gastric cancer type and if the different histological types can affect the overall survival
Author Response
Please see the attachment.

Reviewer 3 Report (New Reviewer)
This is a well written review of an important topic.
Author Response
Thank you very much for your comment.
Kind regards.
Round 2
Reviewer 1 Report (New Reviewer)
Thanks the paper had been improved
This manuscript is a resubmission of an earlier submission. The following is a list of the peer review reports and author responses from that submission.
Round 1
Reviewer 1 Report
The authors sum up the current und past literature regarding treatment of peritoneal metastases of gastric cancer.
I have the following remarks.
1) In order the understand truly the results from literature a structured meta-analysis is necessary.
2) In the palliative treatment section PIPAC, which constitutes the cornerstone of palliative treatment, is hardly mentioned. This section has to be completely updated and must be re-written.
3) The authors refer to laparoscopic HIPEC and NIPS with literaure ref. (2006, and 2012) which are obsolet nowadays. This should be clearly mentioned.
4)Table 1 and 2 are lacking crucial treatment and patient-related parameters (CC-score, PCI-score, perfusion time, open or closed HIPEC and if mentioned incidence of acute kidney injury for Cisplatin-based HIPEC protocols!)
5) Lacking is a treatment algorithm with clearly discussed indication pro or contra CRS and HIPEC (PCI-cut-off? signet-ring and histology in general...)
6) A new chapter/table with ongoing trials should be created.
Reviewer 2 Report
The authors should be congratulated for this extensive review. They did a great job.
Only one comment :
- l.228-231 : there is actually an international delphi study which is on the verge to be published with the consensus regarding the different HIPEC regimen. you may mention it HIPEC Methodology and Regimens: The Need for an Expert Consensus - PubMed (nih.gov)
Reviewer 3 Report
Israel Manzanedoand colleagues conducted a study entitled "Gastric cancer with peritoneal metastases: Current status and prospects for treatment". The topic is interesting, but there are important subjects that I have mentioned below;
The volume of the article is low as a review study, and updated references were not used. On the other hand, the lack of schemes, and figure data is also one of the important shortcomings of this study. The presentation of the tables is not done well. There are many abbreviations in the text that confuse the reader. From the grammatical point of view, the text needs to be revised. All the articles collected in this study are presented in the form of reports and are very brief. In review articles, one should discuss the topic, achievement, and purpose of other articles, taking into account the coherence between the contents. Therefore, I suggest the authors seriously review the study in question.